# Association of Tramadol-Induced Ovarian Damage and Reproductive Dysfunction with Adenosine Triphosphate and the Protective Role of Exogenous ATP Treatment

**DOI:** 10.3390/ph18020216

**Published:** 2025-02-06

**Authors:** Neset Gumusburun, Ilhan Bahri Delibasi, Seval Bulut, Halis Suleyman, Betul Kalkan Yilmaz, Taha Abdulkadir Coban, Ali Sefa Mendil, Zeynep Suleyman

**Affiliations:** 1Department of Gynecology and Obstetrics, Medical Park Tokat Hospital, Tokat 60030, Türkiye; 2Department of Pharmacology, Faculty of Medicine, Erzincan Binali Yildirim University, Erzincan 24100, Türkiye; 3Department of Gynecology and Obstetrics, Faculty of Medicine, Erzincan Binali Yildirim University, Erzincan 24100, Türkiye; 4Department of Biochemistry, Faculty of Medicine, Erzincan Binali Yildirim University, Erzincan 24100, Türkiye; 5Department of Pathology, Faculty of Veterinary Medicine, Erciyes University, Kayseri 38039, Türkiye; 6Department of Internal Medicine Nursing, Faculty of Health Sciences, Erzincan Binali Yildirim University, Erzincan 24100, Türkiye

**Keywords:** tramadol, ATP, oxidative stress, inflammation, reproduction, infertility, rat

## Abstract

**Background:** Tramadol, a weak opioid analgesic agent, is known to induce ovarian damage. Previous studies have held oxidative stress responsible for the adverse effects of tramadol on female reproduction. This study examined the protective effects of ATP against tramadol-induced ovarian damage and reproductive dysfunction in rats. **Methods:** Rats were divided into four groups (*n* = 12); healthy (HG), only ATP (ATPG), only tramadol (TMDG), and ATP + tramadol (ATMG). ATP was injected intraperitoneally at 25 mg/kg. Tramadol at 50 mg/kg was initiated one hour after ATP. The treatment was administered once a day for 14 days. Six rats from each group were euthanized. For two months, the remaining rats were paired with male rats. Rats that failed to give birth during this period were considered infertile. A maternity period was calculated for the rats that were delivered. **Results:** Tramadol caused an increase in malondialdehyde and interleukin-6, and decreased total glutathione, superoxide dismutase, and catalase levels in the ovarian tissue. Furthermore, tramadol disrupted the histological structure of the ovaries, and immunohistochemical staining revealed severe immunopositivity. Tramadol again caused infertility and delayed pregnancy in fertile women. By suppressing biochemical changes, ATP significantly reduced tramadol-induced ovarian damage. Both histopathologically and immunohistochemically, ATP treatment regressed ovarian damage. Additionally, ATP significantly reduced tramadol-induced infertility and maternal delay. **Conclusions:** The results indicate that tramadol-induced oxidative and inflammatory ovarian injury, infertility, and caspase 3 were suppressed by ATP, as demonstrated by our experimental findings.

## 1. Introduction

Tramadol is a synthetic analgesic drug exhibiting opioid and non-opioid properties, with a wide range of applications, mostly in the treatment and relief of moderate to severe pain conditions [1]. Tramadol is often preferred for the treatment of postoperative pain, cancer pain, and chronic neuropathic pain [2]. Tramadol exhibits analgesic properties through a mechanism involving both μ-opioid receptor binding and inhibition of norepinephrine and serotonin reuptake [3]. On the other hand, several disadvantages of tramadol, such as respiratory depression, constipation, and the potential for abuse, limit its use [4]. The use of tramadol is also associated with undesirable side effects such as hepatotoxicity and nephrotoxicity [5]. Tramadol has been reported to cause oxidative and inflammatory damage to the heart and other organs, even at analgesic doses [6]. In the clinic, little information is available on the effects of tramadol on reproductive organs and function in both men and women, although experimental studies have shown that tramadol adversely affects reproductive health in both sexes [7]. In the literature, it has been reported that tramadol causes histopathological damage in the ovaries of female rats, and tramadol-induced reproductive dysfunction and decreased sexual activity may occur [8,9]. Tramadol administration has been shown to cause ovarian damage in rats, and it has been reported that it increases malondialdehyde (MDA) levels in ovarian tissues and causes a decrease in antioxidants such as reduced glutathione (GSH), glutathione peroxidase, and superoxide dismutase (SOD) [10]. Tramadol-induced reproductive organ damage and dysfunction is associated with an increase in proinflammatory cytokines, such as oxidants, interleukin-1β, interleukin-6 (IL-6), tumor necrosis factor-α, nuclear factor-kappa B, and a decrease in antioxidants [11].

When the pathogenesis of tramadol toxicity is analyzed, a decrease in ATP levels accompanying oxidative stress becomes apparent [12]. According to Mousavi et al., adenosine triphosphate (ATP) is depleted in kidney cells with a high level of reactive oxygen species (ROS), lipid peroxidation (LPO), and very low levels of antioxidants [12]. In another study, the attenuating effect of tramadol on mitochondrial respiratory chain enzymes, disruption of mitochondrial membrane potential, and decrease in ATP level were responsible for neurotoxicity [13]. Similarly, Faria et al. proved in vitro that tramadol affected the expression of energy metabolism enzymes and caused ATP depletion [14]. These literature data suggest that the oxidative damage caused by tramadol on ovaries may be due to intracellular ATP depletion. Considering that ATP plays a role in synthesizing ROS-scavenging antioxidants and acts as an energy source for antioxidant synthesis, ATP depletion can be expected to make tissues more vulnerable to oxidative damage [15,16]. These data suggest that ATP may be utilized to prevent tramadol-induced ovarian damage and reproductive dysfunction. Additionally, no information has been found in the literature regarding ATP’s effect on tramadol-induced ovarian damage and reproductive dysfunction. Therefore, this study was designed to investigate the protective effects of ATP on possible tramadol-related ovarian injury and reproductive dysfunction in rats.

## 2. Results

### 2.1. Biochemical Results

MDA, tGSH, SOD, CAT, and IL-6 analysis results.

As shown in Figure 1A and Table 1, the MDA levels of the tramadol-alone group were higher than the healthy rats (*p* < 0.001). Treatment with ATP significantly suppressed the tramadol-induced increase in MDA in the ovarian tissue (*p* < 0.001). MDA levels in the tramadol + ATP group were close to those in the HG (*p* = 0.080).

The tGSH levels obtained from the tramadol group were seen to be decreased according to the HG (*p* < 0.001). ATP inhibited the tramadol-induced decrease in tGSH levels (*p* < 0.001). No significant difference was determined when the tramadol + ATP group was compared with the HG (*p* = 0.272; Figure 1B and Table 1).

Based on the results shown in Figure 1C,D and Table 1, the tramadol group was found to have lower tissue levels of SOD and CAT than the healthy group (*p* < 0.001). As a result of tramadol treatment combined with ATP treatment, tramadol-induced reductions in SOD and CAT activity in the ovaries of rats were significantly inhibited (*p* < 0.001). The SOD data for the tramadol + ATP group and the healthy group were similar (*p* = 0.127).

As illustrated in Figure 1E and Table 1, the IL-6 levels of rats given tramadol alone measured higher than those of healthy rats (*p* < 0.001). ATP treatment significantly suppressed the tramadol-induced increase in IL-6 levels (*p* < 0.001). No significant difference was determined in the IL-6 levels in the ovarian tissue of the tramadol + ATP treatment group compared to the healthy group (*p* = 0.357).

### 2.2. Histopathological Results

The histopathologic examination of ovarian tissues of rats in the HG and ATPG groups demonstrated a normal histologic appearance, as shown in Figure 2A,B. There was, however, a significant degeneration of the zona granulosa cells in the follicles of the ovaries of the rats in the TMDG group due to hemorrhage (Figure 2C). In the follicles of the ovaries of the ATMG group, mild degeneration of zona granulosa cells due to hemorrhage was observed (Figure 2D). An analysis of histopathological data obtained from rat ovarian tissues was presented in Table 2.

### 2.3. Immunohistochemical Results

There were statistically significant differences between the groups regarding immunohistochemical stains labeled with a Caspase 3 antibody (Table 3; *p* < 0.05). As a result of Caspase 3 staining, no significant immunopositivity was observed in the HG or ATPG groups (Figure 3A,B). In the TMDG group, severe immunopositivity was observed (Figure 3C). A mild immunopositivity was observed in the ATMG group that was treated with ATP (Figure 3D). There was immunopositivity in the stroma, corpus luteum, and zona granulosa cells of the follicles.

### 2.4. Fertility Results

ATP’s effects on tramadol-induced infertility and pregnancy delay are shown in Table 4. In the HG and ATPG groups, all six rats (100%) gave birth within the 60-day waiting. In the healthy group, parturition occurred on average 24 days after the male and female rats were placed in the same environment, whereas it took 23.83 days in the ATP group. Furthermore, only two of the six rats (33.3%) in the TMDG group gave birth within the waiting period of 60 days, while the four rats (66.7%) that did not give birth were considered infertile during this period. A delivery time of 34.50 days was recorded in the tramadol group. Five of the six rats (83.3%) receiving Tramadol + ATP treatment gave birth within the 60-day waiting period, while one rat (16.7%) that failed to give birth during this time was considered infertile. In this group, the delivery duration was calculated as 24.20 days.

## 3. Discussion

This study examined the biochemical and histopathological effects of ATP on possible ovarian injury and reproductive dysfunction caused by tramadol in rats. There is wide use of tramadol, a centrally acting synthetic opioid analgesic, in medical practice [3,17]. Its therapeutic effect was mainly ascribed to the activation of μ-opioid receptors and blockade of serotonin and norepinephrine reuptake by synaptosomes [17]. Unlike other opioids, it is also used non-medically to produce euphoria, relaxation, and relieve emotional distress, as well as to stay awake or improve sexual performance since it has both stimulant and antidepressant properties [18]. Its widespread use for both medical and non-medical purposes, especially among young people, along with its low cost, makes tramadol a cause for concern [18,19]. El-Ghawet has demonstrated that tramadol causes ovarian failure, the majority of follicles disappear and are replaced with atretic, cystic follicles, and leading to decreased fertility [2]. The pathogenesis of this ovarian damage has been associated with the development of oxidative stress [20]. There is a large concentration of polyunsaturated fatty acids in biological membranes, which are susceptible to peroxidative attack by oxidants that induce LPO [8,20]. As a result of its interaction with molecular oxygen, tramadol is capable of causing LPO by initiating a series of reactions that lead to the production of free radicals such as superoxide, hydrogen peroxide, hydroxyl, and peroxynitrite [8]. As a result of our experiments, we found an increase in MDA levels in the ovarian tissue of tramadol-treated animals according to both healthy and ATP control groups. It is well known that the LPO reaction produces a wide variety of oxidation products, and while the primary products produced are lipid hydroperoxide, MDA is one of the many aldehydes that occur as secondary products [21,22]. The most prominent and predominant product of polyunsaturated fatty acid peroxidation is MDA [23]. MDA is a widely used indicator of oxidative stress [24]. An increase in MDA levels in a tissue indicates an increase in ROS. The structure and function of the membrane can be greatly impacted as a result of the damage caused by MDA, which is itself toxic [24]. It has been demonstrated, in parallel with our study, that ovarian MDA levels are increased in female albino rats administered 30 and 60 mg/kg of tramadol for 8 weeks at both doses in a study by El-Twab [10]. Furthermore, Hindawy et al. demonstrated an increase in blood serum MDA levels in both female and male albino rats when tramadol was administered at 50 mg/kg/day for four weeks [25].

As a result of our biochemical experiments, we demonstrated that tGSH, SOD, and CAT activities in ovarian samples of tramadol-treated animals differed significantly from those of healthy animals and those treated with ATP alone. In essence, oxidative stress refers to the shift in the ratio of pro-oxidants to antioxidants in favor of pro-oxidants, resulting in potential damage and characterized by the release of ROS [26]. To prevent damage to cells, tissues, and organs, excessive levels of ROS are neutralized by non-enzymatic and enzymatic antioxidants [27]. An endogenous non-enzymatic antioxidant tripeptide, GSH is composed of three amino acids: glutamate, glycine, and cysteine [28]. Although GSH has direct antioxidant effects, it is primarily responsible for performing its antioxidant function through glutathione peroxidase-catalyzed reactions in which it reduces and scavenges H_2_O_2_, organic peroxides, such as lipid hydroperoxides, and peroxynitrites [29]. GSH contains an active thiol group that can readily be oxidized and dehydrogenated, thereby eliminating superoxide radicals and supplying electrons to enzymes, such as glutathione peroxidase, that can reduce H_2_O_2_ to H_2_O [30]. Among its functions are maintaining intracellular redox balance, reducing oxidative damage, and inhibiting apoptosis [30]. As an enzymatic antioxidant, SOD converts superoxide radicals into H_2_O_2_, while CAT, another enzymatic antioxidant, hydrolyzes H_2_O_2_ molecules into harmless compounds [31]. The literature indicates that the decreased levels of enzymatic and non-enzymatic antioxidants are a result of the excessive consumption of antioxidants during the neutralization of ROS support our experimental findings [32]. For example, Paulis et al. reported that ovarian tGSH, SOD, and CAT levels decreased when female albino rats were administered 40 and 80 mg/kg/day tramadol for eight weeks at both doses [20]. El-Twab also found that ovarian tGSH and SOD levels decreased at both doses of tramadol administered to female albino rats for eight weeks at 30 and 60 mg/kg/day [10]. Our experimental results and the existing literature indicate that tramadol administration disrupts the redox balance in ovarian tissue in favor of oxidants, antioxidants decrease due to excessive consumption, and consequently, oxidative stress develops. In addition, it was shown that there was no difference between the healthy group and the ATP group in terms of oxidant and antioxidant data. These data indicate that redox equilibrium is present in ovarian tissues belonging to the ATP group.

In addition, tramadol-treated rats had significantly higher levels of IL-6 in their ovarian tissue than healthy or ATP alone-treated rats. IL-6 is a single-chain protein produced by T cells, B cells, monocytes, fibroblasts, and other types of cells [33]. As a member of the proinflammatory cytokine family, IL-6 plays an active role in the pathogenesis of inflammation by inducing the expression of various proteins that play a role in acute inflammation [34,35]. In the literature, there is little evidence that tramadol administration increases IL-6 levels in the ovary. However, Adelakun et al. demonstrated that tramadol administration to adult male rats at 50 mg/kg/day for 8 weeks increased in liver and kidney IL-6 levels [11]. As a result of our experiments and information obtained from the literature, tramadol may be associated with ovarian inflammation.

In our study, we investigated whether exogenous ATP treatment can protect the ovarian tissues of rats from tramadol damage. There is no consensus on the mechanism of action of exogenous ATP treatment. Contrary to the view that ATP cannot enter and act inside the cell, it was shown that labeled ATP could enter muscle cells [36]. For ATP therapy, the rapid degradation of ATP has also been a source of concern; conversely, it has been suggested that AMP, the breakdown product of ATP, mediates the protective effects of ATP. It has been argued that AMP itself or the breakdown products of AMP can be transported across the plasma membrane into cells where it can be synthesized into ATP [37]. In the literature, it has been reported that ATP regulates many cellular functions through purinergic P_2_ receptors [38]. The results of our study revealed that exogen ATP treatment was effective. It was found that the increases in MDA and IL-6 levels, and decreases in tGSH, SOD, and CAT levels induced by tramadol in rat ovaries were significantly suppressed after tramadol + ATP treatment. ATP contains a nitrogenous base (adenine), a ribose sugar, and three phosphate groups [39]. ATP plays a critical role in sustaining cell viability and is synthesized in mitochondria through oxidative phosphorylation, which involves transferring electrons from energy substrates to oxygen [40]. It is the mitochondria that are the primary source of intracellular ROS generation, and therefore ROS generation and oxidative stress may impair mitochondrial function [41]. Tramadol is also known to significantly reduce mitochondrial activity and energy metabolism. [12,42,43]. Furthermore, Mohammednejad et al. suggest that excessive ROS production caused by tramadol administration results in mitochondrial dysfunction and damage through defects of complex II and membrane permeability transition pores, collapse of mitochondrial membrane potentials, and swelling of mitochondria [44]. As a result of mitochondrial dysfunction, ATP production and ATP depletion are reduced, mitochondrial respiration is impaired, and ROS are produced in an increased amount, which activates harmful cellular pathways [40]. The intracellular level of ATP plays an important role in determining how cells die, and in the absence of ATP, necrosis is preferred over apoptosis [45]. All these changes may be reversed by the addition of extracellular ATP, despite a lack of data on intravenous ATP administration.

The histopathologic and immunohistochemical findings of this study supported the biochemical results. According to our histopathological findings, the tramadol group had very severe stromal and follicular hemorrhage, severe zona granulosa cell degeneration in the follicles, and very severe edema in the ovarian tissue. In agreement with our histopathological findings, Mohamed et al. reported that ovarian follicles of albino Wistar rats treated with tramadol lost their normal structure and degenerated [46]. Similarly, Hindawy et al. administered oral tramadol for four weeks to 50 albino Wistar rats and observed degeneration of the follicles [25]. A study conducted by Elhomosany, however, reported the development of follicular degeneration in granulosa cells with pyknosis in the nucleus and dissolution in the cytoplasm in 40 rats that were administered tramadol HCL for 60 days [47]. Treatment with ATP, however, significantly suppressed the very severe stromal and follicular hemorrhage, the degeneration of zone granulosa cells in follicles, as well as the very severe edema induced by tramadol in our study. These findings confirmed the histopathologic findings of Ozer et al. demonstrating that ATP prevented the ovaries from injury caused by 5-fluorouracil [39]. According to our immunohistochemistry study, carried out using the Caspase 3 antibody, stromal cells, corpus luteum cells, and zona granulosa cells of follicles in the tramadol group, showed high levels of immunoreactivity. Our immunohistochemical findings are consistent with those of Elhomosany’s study, in which the ovarian stroma, corpus luteum, and follicles of tramadol-treated rats were highly immunopositive for caspase 3 [47]. ATP treatment attenuated this immune positivity, according to our experimental results.

Despite studies reporting that tramadol causes histopathologic damage to the ovaries of female rats that may lead to reproductive dysfunction [48], there is no information available regarding infertility and pregnancy delay. According to our experimental findings, tramadol significantly increased the incidence of infertility (66.7%) within a 60-day waiting period. The use of tramadol also resulted in a prolongation of maternity (34.5 days). On the other hand, ATP significantly inhibited tramadol-induced infertility (16.7%) and reduced the duration of maternity (24.2 days).

## 4. Materials and Methods

### 4.1. Animals

Forty-eight (*n* = 48) female and eight (*n* = 8) male albino Wistar rats (9–10 weeks old, 270 and 285 g) were used for the study. All animals were obtained from Erzincan Binali Yildirim University Experimental Animals Application and Research Center. In our study, the ARRIVE guide was taken into consideration. In a laboratory environment, rats were housed in groups of twelve in conventional cages at 22 °C with 12 h light/dark cycles and 30–70% humidity. Rats were provided with ad libitum access to standard pellet feed and water. It was conducted in accordance with local legislation and institutional requirements.

### 4.2. Chemicals

Thiopental sodium was obtained from IE Ulagay (Istanbul, Türkiye), ATP was from Zdorove Narodu (Kharkiv, Ukraine), and Tramadol HCl was from Abdi Ibrahim Ilaç Sanayi (Istanbul, Türkiye).

### 4.3. Experimental Groups

Throughout the experiment, 48 female rats were allocated to four groups: a healthy group (HG), an ATP-alone group (ATPG), a tramadol-alone group (TMDG), and ATP + tramadol group (ATMG).

### 4.4. Experimental Procedure

In the ATPG and ATMG groups, ATP 25 mg/kg was given intraperitoneally (IP) [49]. A similar amount of pure water was applied to the HG and TMDG groups by IP. One hour after giving ATP and pure water, the ATMG and TMDG groups received tramadol 50 mg/kg by oral gavage. This process was replicated once a day for 14 days [50]. Six rats from every group were euthanized with high doses of thiopental sodium (50 mg/kg) and the ovaries were excised at the end of this period. MDA, tGSH, SOD, CAT, and IL-6 levels were measured in ovarian tissues. Histopathological examination was performed on another part of the tissues. Six female rats from each group were housed in the laboratory with two adult male rats per group for 60 days. The reproductive period was determined as 60 days with reference to previous studies considering the possibility that drugs that adversely affect reproductive functions may cause a delay in conception [51]. During this period, rats found to be pregnant by physical examination were moved to separate cages where they could stay alone. Those rats that did not give birth within the 60 days were considered infertile. The maternity period was determined by subtracting the standard pregnancy period (average of 21–23 days) from the time between when the female rats joined with the male rats and the day of birth (A) (A-22 = maternity period).

### 4.5. Biochemical Analyses

#### Preparation of Samples

After washing the ovarian tissue with physiological saline, it was ground into powder in liquid nitrogen and homogenized. Clear filtrate was used for MDA, GSH, SOD, and CAT analyses.

### 4.6. MDA, tGSH, SOD, CAT, Protein, and IL-6 Levels Analysis in Ovarian Tissue

A commercial ELISA kit was used to assay the levels of MDA, GSH, and SOD and each was performed as described in the kit instructions (MDA, tGSH, and SOD. Cat No: 10009055, 703002, 706002, respectively, Cayman Chemical Company, Ann Arbor, MI, USA). The measurement of CAT was performed according to the method recommended by Goth [52]. Using Bradford’s method, protein concentration was measured by spectrophotometry at 595 nm [53]. IL-6 levels were determined using an available ELISA kit (Hangzhou Eastbiopharm Co., Ltd., Hangzhou, China).

### 4.7. Histopathological Examination

A necropsy was performed on the rats, and the ovarian tissues were fixed in a 10% formalin solution. After being processed with routine alcohol–xylol, the tissues were placed in paraffin blocks. Then, 5 µm sections were obtained for histopathologic evaluation. The 5-µm sections taken on poly-lysine slides were stained with Hematoxylin–Eosin (H&E), and the tissues were evaluated by a pathologist unaware of the treatment protocol with a light microscope (Olympus BX 51, Tokyo, Japan) and images were taken with a digital camera (Olympus DP 71, Tokyo, Japan). Six random sections from each ovary were used to determine the level of histopathological damage. Each tissue section was evaluated for stromal hemorrhage, follicular hemorrhage, follicular degeneration, and edema. To determine the level of damage, the damage square was measured for each parameter and the percentage ratio to the total square of the histological section was calculated. For stromal hemorrhage and edema, <20% was considered mild damage, 20–40% moderate damage, 40–60% severe damage, and >60% very severe damage. For follicular hemorrhage and degeneration, <25% mild damage, 25–50% moderate damage, 50–75% severe damage, and >75% very severe damage were evaluated. Histopathological evaluation was determined by modifying the histopathological evaluation criteria of Karateke et al. [54].

### 4.8. Immunohistochemical Method

5 μm sections taken on poly-lysine slides were treated with xylol and alcohol, washed with PBS (Phosphate-Buffered Saline), and then kept in 3% hydrogen peroxide (H_2_O_2_) for 10 min to inactivate endogenous peroxidase. The tissues were treated with antigen retrieval solution at 500 watts for 2 × 5 min to release the antigen from the tissues. The tissues were then washed with PBS and incubated with cleaved caspase-3 (Elabscience, Cat no. E-AB-30004) primary antibodies at 1/200 dilution ratio at +4 °C overnight. Second, a Large Volume Detection System: anti-polyvalent, HRP (Thermo Fischer, LabVision Corporation, Fremont, CA, USA, Catalog no. TP-125-HL) was used by the manufacturer’s recommendations. 3,3′-Diaminobenzidine was used as a chromogen. The slides were counterstained with Mayer’s Hematoxylin, covered with Entellan, and examined under a light microscope. During the examination, immunopositivity was evaluated semi-quantitatively as absent (0), mild (1), moderate (2), and severe (3).

### 4.9. Statistical Analysis

The statistical analyses were conducted using the IBM SPSS Statistical Program for Windows (IBM Corp., Version 27.0, released in 2020, Armonk, NY, USA). The figures were created using the GraphPad Prism program (GraphPad Software, Version 8.0.1, released in 2018, San Diego, CA, USA). Numerical quantitative data were expressed as “Mean ± Standard Error of the Mean”. In the case of continuous variables, the Shapiro–Wilk test was used to determine whether the distribution was normal. A one-way ANOVA test was used to analyze normally distributed variables. According to the results of Levene’s test, if the homogeneity of variances was ensured, Tukey’s HSD was used as a post hoc test to determine the group causing the difference, and if not, the Games–Howell test was used. The result of immunohistochemical qualitative data were represented as “Mean ± Standard Error of the Mean” and “Median (Minimum–Maximum)” and the Kruskal–Wallis test followed by the Dunn–Bonferroni–test was used in the analysis. A value of *p* < 0.05 was accepted to be statistically significant.

## 5. Conclusions

The findings of the current study revealed that tramadol administration caused oxidative and inflammatory damage in rat ovaries by encouraging the production of oxidative and proinflammatory cytokines and reducing antioxidant stores. Tramadol-induced oxidative and inflammatory ovarian damage has been demonstrated biochemically, histopathologically, and immunohistochemically. Histopathological evidence showed that tramadol induced severe degeneration of the zona granulosa cells of the ovarian follicles of rats, while immunohistochemical staining labeled with Caspase 3 antibodies revealed severe immunopositivity. Tramadol treatment resulted in infertility in some of the animals, and delayed maternity periods were observed in those animals who did not experience infertility. ATP reduced the ovarian damage caused by tramadol by suppressing the increase in levels of the oxidant MDA and proinflammatory IL-6, and the decrease in levels of antioxidants tGSH, SOD, and CAT. A combination of histopathological and immunohistochemical analysis demonstrated that ATP treatment reduced ovarian damage to a mild degree. Furthermore, ATP significantly reduced the development of tramadol-induced infertility and delayed pregnancy. Tramadol-induced oxidative and inflammatory damage to the ovary may be treated with ATP according to our experimental results. Women of reproductive age should use tramadol with caution.

### Limitations

In order to clarify the mechanism, further molecular studies are required.

## Figures and Tables

**Figure 1 pharmaceuticals-18-00216-f001:**
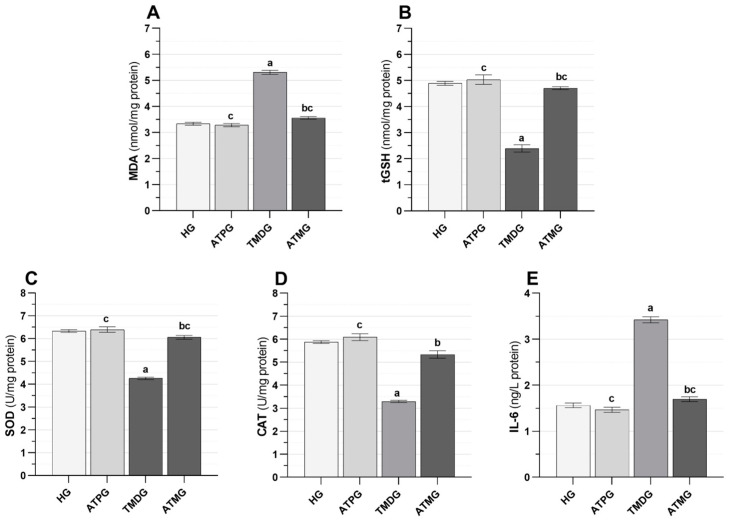
Effects of adenosine triphosphate and tramadol on MDA (**A**), tGSH (**B**), SOD (**C**), CAT (**D**), and IL-6 (**E**) levels in rat ovarian tissue. a: *p* < 0.001 vs. HG; b: *p* < 0.001 vs. TMDG; c: *p* > 0.05 vs. HG; HG: healthy group; ATPG: adenosine triphosphate alone group; TMDG: tramadol alone group; ATMG: adenosine triphosphate + tramadol group; MDA: malondialdehyde; tGSH: total glutathione; SOD: superoxide dismutase; CAT: catalase; IL-6: interleukin-6. Statistical analyses were performed using the one-way ANOVA test—Tukey’s HSD or Games–Howell tests.

**Figure 2 pharmaceuticals-18-00216-f002:**
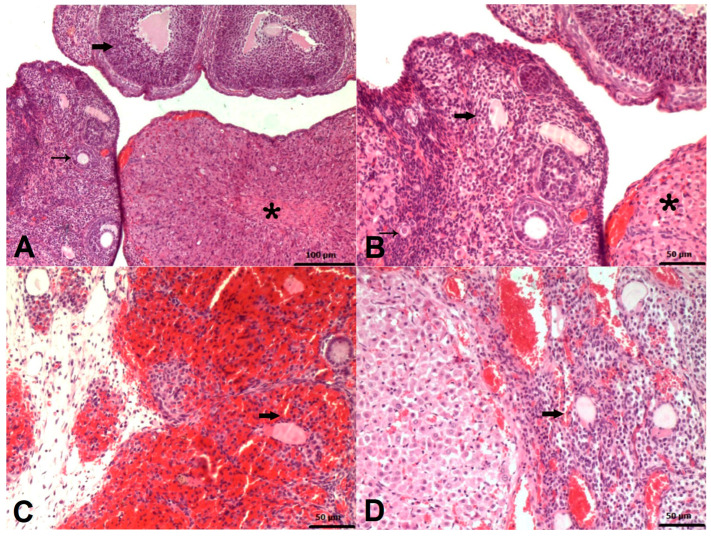
(**A**) Histologic appearance of ovarian tissue of HG. Corpus luteum (*), Primary follicle (thin arrow), Secondary follicle (thick arrow), (HxE); (**B**) Histologic appearance of ovarian tissue of ATPG. Corpus luteum (*), Primordial follicle (thin arrow), Secondary follicle (thick arrow), (HxE); (**C**) Histologic appearance of ovarian tissue of TMDG. Severe degeneration of zona granulosa cells (arrow) due to hemorrhage in the follicles of ovaries of TMDG (HxE); (**D**) Histologic appearance of ovarian tissue of ATMG. Mild degeneration of zona granulosa cells (arrow) due to hemorrhage in the follicles of the ovaries of ATMG (HxE).

**Figure 3 pharmaceuticals-18-00216-f003:**
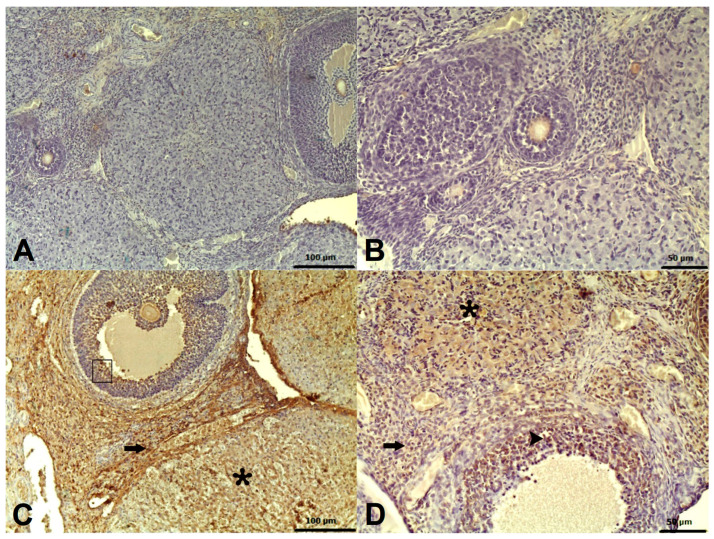
(**A**) Caspase 3 immune negativity (IHC) of ovarian tissue from HG; (**B**) Caspase 3 immune negativity (IHC) of ovarian tissue from ATPG; (**C**) Severe caspase 3 immunopositivity (IHC) in corpus luteum (*), stroma (arrow), follicles (□) of ovarian tissue of TMDG; (**D**) Mild caspase 3 immunopositivity (IHC) in corpus luteum (*), stroma (arrow), zona granulosa cells in follicles (arrowhead) of ovarian tissue of ATMG.

**Table 1 pharmaceuticals-18-00216-t001:** Effects of adenosine triphosphate and tramadol on oxidant, antioxidant, and proinflammatory cytokine levels in rat ovarian tissue.

Biochemical Variables		MDA * (nmol/mg Protein)	tGSH ** (nmol/mg Protein)	SOD * (U/mg Protein)	CAT * (U/mg Protein)	IL-6 (ng/L Protein)
	(Mean ± Standard Error of the Mean)					
**Groups (*n* = 6/each group)**	**HG**	3.34 ± 0.06	4.89 ± 0.08	6.33 ± 0.06	5.88 ± 0.06	1.56 ± 0.05
	**ATPG**	3.28 ± 0.06	5.03 ± 0.18	6.40 ± 0.12	6.09 ± 0.15	1.47 ± 0.06
	**TMDG**	5.31 ± 0.08	2.40 ± 0.14	4.26 ± 0.05	3.29 ± 0.05	3.42 ± 0.06
	**ATMG**	3.56 ± 0.05	4.71 ± 0.06	6.06 ± 0.09	5.34 ± 0.16	1.70 ± 0.05
**Group comparison** ***p* values**	**HG vs. ATPG**	0.936	0.890	0.950	0.583	0.633
	**HG vs. TMDG**	<0.001	<0.001	<0.001	<0.001	<0.001
	**HG vs. ATMG**	0.080	0.272	0.127	0.019	0.357
	**ATPG vs. TMDG**	<0.001	<0.001	<0.001	<0.001	<0.001
	**ATPG vs. ATMG**	0.024	0.397	0.045	0.001	0.042
	**TMDG vs. ATMG**	<0.001	<0.001	<0.001	<0.001	<0.001
**F (3, 20)**		238.701	102.659	144.115	120.014	275.024
***p* values**		<0.001	<0.001	<0.001	<0.001	<0.001

Statistical analyses were performed using the one-way ANOVA test followed by Tukey’s (*) or Games–Howell (**) post hoc tests. *p* < 0.05 was determined as statistical significance. HG: healthy group; ATPG: adenosine triphosphate alone group; TMDG: tramadol alone group; ATMG: adenosine triphosphate + tramadol group; MDA: malondialdehyde; tGSH: total glutathione; SOD: superoxide dismutase; CAT: catalase; IL-6: interleukin-6.

**Table 2 pharmaceuticals-18-00216-t002:** Histopathological grading analysis of rat ovaries.

Histopathological Variables	Groups	Stromal Hemorrhage	Follicular Hemorrhage	Follicular Degeneration	Edema
		Mean ± Standard Error of the Mean			
**Groups (*n* = 6/each group)**	**HG**	0	0	0	0
	**ATPG**	0	0	0	0
	**TMDG**	64.67 ± 1.58	75.83 ± 1.54	65.83 ± 1.25	69.33 ± 1.15
	**ATMG**	13.17 ± 2.93	20.67 ± 0.67	16.50 ± 0.54	11.00 ± 0.73
**Group comparison *p* values**	**HG vs. ATPG**	-	-	-	-
	**HG vs. TMDG**	<0.001	<0.001	<0.001	<0.001
	**HG vs. ATMG**	<0.001	<0.001	<0.001	<0.001
	**ATPG vs. TMDG**	<0.001	<0.001	<0.001	<0.001
	**ATPG vs. ATMG**	<0.001	<0.001	<0.001	<0.001
	**TMDG vs. ATMG**	<0.001	<0.001	<0.001	<0.001
**F (3, 20)**		961.568	1829.581	1160.764	2395.205
***p*** **values**		<0.001	<0.001	<0.001	<0.001

Statistical analyses were performed using the one-way ANOVA test followed by a post hoc Games–Howell test. *p* < 0.05 was determined as statistical significance. HG: healthy group; ATPG: adenosine triphosphate alone group; TMDG: tramadol alone group; ATMG: adenosine triphosphate + tramadol group.

**Table 3 pharmaceuticals-18-00216-t003:** Analysis of immunohistochemical grading data from rat ovarian tissue.

**Caspase 3 (Immunohistochemical grading)**	**Groups (*n* = 6/each group)**				**H**	***p* Value**
	**HG**	**ATPG**	**TMDG**	**ATMG**		
	**Mean ± Standard Error of the Mean Median (minimum–maximum)**					
	0.00 ± 0.00 0 (0–0)	0.00 ± 0.00 0 (0–0)	2.83 ± 0.17 3 (2.75–3)	1.00 ± 0.26 1 (1–1)	220.913	<0.001
	**Group comparison/*p* values**					
	**HG vs.**			**ATPG vs** **.**		**TMDG vs** **.**
	**ATPG**	**TMDG**	**ATMG**	**TMDG**	**ATMG**	**ATMG**
	1.000	<0.001	0.247	<0.001	0.247	0.395

Histopathologic grading: 0-none, 1-mild, 2-moderate, 3-severe. Statistical analyses were performed using the Kruskal–Wallis test followed by the post hoc Dunn–Bonferroni test. *p* < 0.05 was determined as the statistical significance. HG: healthy group; ATPG: adenosine triphosphate alone group; TMDG: tramadol alone group; ATMG: adenosine triphosphate + tramadol group.

**Table 4 pharmaceuticals-18-00216-t004:** Effects of ATP on tramadol-induced infertility.

Groups	*n*		Fertile Animals			Infertile Animals			Reproductive Process (A) (days)		Delay in Maternity (A-22)	
			*n*		%	*n*		%	Mean ± Standard Error of the Mean			
**HG**	6		6		100	-		-	24.00 ± 0.37		2.00 ± 0.37	
**ATPG**	6		6		100	-		-	23.83 ± 0.48		1.83 ± 0.48	
**TMDG**	6		2		33.3	4		66.7	34.50 ± 1.50		13.00 ± 1.00	
**ATMG**	6		5		83.3	1		16.7	24.20 ± 0.37		2.20 ± 0.37	
**F (3, 20)**											54.501	
***p* value**											<0.001	
	**Group comparison *p* values**											
**HG vs. ATPG**		**HG vs. TMDG**		**HG vs.ATMG**			**ATPG vs. TMDG**			**ATPG vs. ATMG**		**TMDG vs. ATMG**
0.993		<0.001		0.990			<0.001			0.945		<0.001

Statistical analyses were performed using the one-way ANOVA test followed by Tukey’s post hoc test. *p* < 0.05 was determined as statistical significance. HG: healthy group; ATPG: adenosine triphosphate alone group; TMDG: tramadol alone group; ATMG: adenosine triphosphate + tramadol group; *n*: number of animals.

## Data Availability

The data presented in this study are available on request from the corresponding author due to ethical reasons.

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
