# Peer review of "Association of Tramadol-Induced Ovarian Damage and Reproductive Dysfunction with Adenosine Triphosphate and the Protective Role of Exogenous ATP Treatment"

_pharmaceuticals, 2025, doi:10.3390/ph18020216_

Round 1
Reviewer 1 Report
Comments and Suggestions for Authors
The overall goal of this study was to assess the protective effects of ATP on possible tramadole-related ovarian injury and reproductive dysfunction in rats.
The abstract should provide a background about tramadole.
The title I think should be revised because it sounded not reflective of the overall goal of the study.
The introduction is very short. This should be further substantiated. I also suggest to break down the introduction into several paragraphs. In the introduction, there seems to be so many pieces going on and this can be further improved by breaking down into multiple paragraphs. Also, the topic regarding ATP just suddenly appeared in the introduction. I feel like it appears like it is out of place and should be further discussed. Please emphasize on the topic of ATP and its role in relation to tramadole.
Table 2 looks distorted. Table 3 is the same as well.
The references font sizes and style seemed to be incorrect.
I think the study is interesting and could provide good insights into the protective effects of ATP on possible tramadole related ovarian injury but it needs a lot of work before being published.
Comments on the Quality of English Language
The manuscript requires some English language copyediting improvement.
Author Response
Reviewer 1
-The overall goal of this study was to assess the protective effects of ATP on possible tramadole-related ovarian injury and reproductive dysfunction in rats.
-The abstract should provide a background about tramadole.
Response: Added information about tramadol to the abstract.
-The title I think should be revised because it sounded not reflective of the overall goal of the study.
Response: Title revised.
-The introduction is very short. This should be further substantiated. I also suggest to break down the introduction into several paragraphs. In the introduction, there seems to be so many pieces going on and this can be further improved by breaking down into multiple paragraphs. Also, the topic regarding ATP just suddenly appeared in the introduction. I feel like it appears like it is out of place and should be further discussed. Please emphasize on the topic of ATP and its role in relation to tramadole.
Response: The introduction section has been edited and expanded based on your suggestion.
-Table 2 looks distorted. Table 3 is the same as well.
Response: Tables were arranged.
-The references font sizes and style seemed to be incorrect.
Response: References were arranged.
-I think the study is interesting and could provide good insights into the protective effects of ATP on possible tramadole related ovarian injury but it needs a lot of work before being published.
Thank you for your careful review.

Reviewer 2 Report
Comments and Suggestions for Authors
This study examined the protective effects of adenosine triphosphate (ATP) against tramadol-induced ovarian damage and reproductive dysfunction in rats. Tramadol exposure increased oxidative stress markers, reduced antioxidant levels, and caused significant ovarian tissue damage, infertility, and delayed pregnancies. ATP treatment effectively mitigated these effects by reducing oxidative and inflammatory damage, restoring antioxidants, and improving fertility outcomes.
General impression
I am of the opinion that the authors have conducted a comprehensive investigation and communicated their results proficiently. The writing demonstrates a skillful and systematic approach with a seamless alignment between the utilized methods and the study's objectives. The detailed procedures provide a concise and thorough explanation, maintaining overall clarity and coherence throughout
Title
The title is clear and informative, effectively conveying the study's main focus.
Abstract
The abstract effectively communicates the structure, methodologies, and principal discoveries of the study.
Introduction
The introduction is thorough, with sufficient background information on tramadol’s effects and the rationale for the study.
Citations are well-integrated, but the flow between the paragraphs could be improved for coherence.
The novelty of the study is stated but could be more explicitly emphasized, particularly in comparison to previous research on ATP's protective effects.
Methods
The methodology is detailed, including animal groupings, drug dosages, and biochemical analyses.
Results
The results are presented clearly with appropriate use of tables and figures.
Discussion
The discussion effectively interprets the results and connects them to existing literature.
References
References are comprehensive and relevant, though some are slightly dated. Including more recent studies would improve the article's impact.
Author Response
Reviewer 2
I am of the opinion that the authors have conducted a comprehensive investigation and communicated their results proficiently. The writing demonstrates a skillful and systematic approach with a seamless alignment between the utilized methods and the study's objectives. The detailed procedures provide a concise and thorough explanation, maintaining overall clarity and coherence throughout
Title
The title is clear and informative, effectively conveying the study's main focus.
Abstract
The abstract effectively communicates the structure, methodologies, and principal discoveries of the study.
Response: Thank you
Introduction
The introduction is thorough, with sufficient background information on tramadol’s effects and the rationale for the study.
Citations are well-integrated, but the flow between the paragraphs could be improved for coherence.
The novelty of the study is stated but could be more explicitly emphasized, particularly in comparison to previous research on ATP's protective effects.
Response: Added.
Methods
The methodology is detailed, including animal groupings, drug dosages, and biochemical analyses.
Response: Thank you
Results
The results are presented clearly with appropriate use of tables and figures.
Response: Thank you
Discussion
The discussion effectively interprets the results and connects them to existing literature.
Response: Thank you
References
References are comprehensive and relevant, though some are slightly dated. Including more recent studies would improve the article's impact.
Response: Newer references were used when making additions to the article.

Reviewer 3 Report
Comments and Suggestions for Authors
The manuscript entitled "The Association of Tramadol-Induced Oxidative Ovarian Damage and Reproductive Dysfunction with Adenosine Triphosphate" focuses the reader's attention on the action of tramadol on female reproductive system. In general, it is an interesting manuscript, 4 different groups of animals were investigated. However, I have multiple questions and recommendations for the authors.
1. In their experiments authors used groups which contained 6 female and 2 male rats. In the tramadol-treated group they observed lower levels of pregnancies in comparison to the other groups. However, their conclusions based on that experiments are doubtful that tramadol lead to the infertility. First of all, the number of aminals in the group is small. I recommend for the author to replicate that experiment with tramadol at least twice. Moreover, due to the limited number of animals in the group the abscence of offspring may be caused due to the various reasons: a) females from that group do not allow to the males to perform coitus (personal enmity), to avoid that factor you should change males in the group b) coitus occurs but no pups birth were observed due to the abortus
2. How did you checked the fact of coitus and pregnancy beggining?
3. Why you cosidered 60days interval to concluded that females are inferile? It is your own idea or it is commonly used time for such experiments?
4. Discuss why there is nto differences between HG and ATP groups.
5. Did the exogenous ATP entered into the ovaries? Or there are some ATP-mediated mechanisms of saving ovaries from tramadol? Did ypu measured levels of ATP in ovaries from different groups?
6. Why you selected only described dose of ATP? Maybe increasing of ATP will result in better tramadol-protective properties?
7. It will be interesting to see the effects of another antioxidants, for example ascorbic acid + tramadol group.
8. In figure 2 you note that normal histology is given. It is better to avoid in such cases utilization of word "normal", just, please. indicate which group was analyzed.
9. Section 4.9, statistics. You mention that Kruskal-Wallis followed with Mann-Whitney analysis was used. However, if you perform comparative analysis of multiple groups using Mann-Whitney you should use Bonferroni correction.
10. Lines 329 and 343. What do you mean "semi-quantitively"? You should perform analysis in concrete data. For example, you should measure the square of haemorraghy and calculate the percentage to the total square of histological section. The same for other parameters.
Author Response
Reviewer 3
The manuscript entitled "The Association of Tramadol-Induced Oxidative Ovarian Damage and Reproductive Dysfunction with Adenosine Triphosphate" focuses the reader's attention on the action of tramadol on female reproductive system. In general, it is an interesting manuscript, 4 different groups of animals were investigated. However, I have multiple questions and recommendations for the authors.
- In their experiments authors used groups which contained 6 female and 2 male rats. In the tramadol-treated group they observed lower levels of pregnancies in comparison to the other groups. However, their conclusions based on that experiments are doubtful that tramadol lead to the infertility. First of all, the number of aminals in the group is small. I recommend for the author to replicate that experiment with tramadol at least twice. Moreover, due to the limited number of animals in the group the abscence of offspring may be caused due to the various reasons: a) females from that group do not allow to the males to perform coitus (personal enmity), to avoid that factor you should change males in the group b) coitus occurs but no pups birth were observed due to the abortus
Response: You are right about your concerns. Repeating the experiments will definitely contribute to our study results. However, due to the lack of a funder for the study and the expenses being covered by the authors, in other words, due to financial reasons, we could not repeat the experiments. We kept the number of animals at a minimum level based on our previous studies. We can increase this number in the future studies. You are right about the two concerns you mentioned that could cause the number of offspring to be low. We could not think about this issue. Since we did not have the chance to repeat the experiments, we cannot obtain new data. Your valuable contributions will guide our future studies.
- How did you checked the fact of coitus and pregnancy beggining?
Response: Pregnancy status was determined by physical examination. and then female rats were placed in separate cages. This information was added to the experimental procedure section.
- Why you cosidered 60days interval to concluded that females are inferile? It is your own idea or it is commonly used time for such experiments?
Response: The 60-day period was determined based on previous studies and references are included in the experimental procedure text below.
‘The reproductive period was determined as 60 days with reference to previous studies against the possibility that drugs that adversely affect reproductive functions may cause a delay in conception 52. During this period, rats found to be pregnant by physical examination were moved to separate cages where they could stay alone.’
- Discuss why there is nto differences between HG and ATP groups.
Response: Information on the comparison of HG and ATP groups was added to the Discussion section.
- Did the exogenous ATP entered into the ovaries? Or there are some ATP-mediated mechanisms of saving ovaries from tramadol? Did ypu measured levels of ATP in ovaries from different groups?
Response: The following text on possible protective mechanisms of exogenous ATP on tissues has been added to the discussion section. However, unfortunately, ovarian ATP levels were not measured in this study.
‘In our study, we investigated whether exogenous ATP treatment can protect the ovarian tissues of rats from tramadol damage. There is no consensus on the mechanism of action of exogenous ATP treatment. Contrary to the view that ATP cannot enter and act inside the cell, it was shown that labeled ATP could enter muscle cells (Chaudry, 1982). For ATP therapy, the rapid degradation of ATP has also been a source of concern; conversely, it has been suggested that AMP, the breakdown product of ATP, mediates the protective effects of ATP. It has been argued that AMP itself or the breakdown products of AMP can be transported across the plasma membrane into cells where it can be synthesized into ATP (Mandel et al., 1988). In the literature, it has been reported that ATP regulates many cellular functions through purinergic P2 receptors (Patel et al., 2018). The results of our study revealed that exogen ATP treatment was effective. It was found that the increases in MDA and IL-6 levels and decreases in tGSH, SOD and CAT levels induced by tramadol in rat ovaries were significantly suppressed after tramadol+ATP treatment.’
- Why you selected only described dose of ATP? Maybe increasing of ATP will result in better tramadol-protective properties?
Response: Before, ATP treatment was studied and published by our team in different organs intraperitoneally. The dose of this study was determined by referencing previous studies and we used the dose that we thought was effective. Different doses could have been studied, you are right. The fact that the expenses of the study were covered by the authors was a limiting factor for us.
- It will be interesting to see the effects of another antioxidants, for example ascorbic acid + tramadol group.
Response: Yes, you are right, it could be interesting. This study design did not include a second product whose protective effect was being investigated. We did not have the opportunity to add ascorbic acid at this stage. We will definitely evaluate your suggestion in a separate study.
- In figure 2 you note that normal histology is given. It is better to avoid in such cases utilization of word "normal", just, please. indicate which group was analyzed.
Response: Figure description has been changed based on your suggestion.
- Section 4.9, statistics. You mention that Kruskal-Wallis followed with Mann-Whitney analysis was used. However, if you perform comparative analysis of multiple groups using Mann-Whitney you should use Bonferroni correction.
Response: Thank you for your attention. There is a typo. The Dunn Bonferroni test was used post hoc. The typo has been corrected.
- Lines 329 and 343. What do you mean "semi-quantitively"? You should perform analysis in concrete data. For example, you should measure the square of haemorraghy and calculate the percentage to the total square of histological section. The same for other parameters.
Response: By semi-quantitative we mean ordinal data. Histopathological data were scored between 0-4 according to the damage criteria. Since these data were accepted as ordinal data (semiquantitative), the analysis was performed with the Kruskal-Wallis test, which is a nonparametric test. Scoring criteria and relevant references were added to the material method section.

Round 2
Reviewer 1 Report
Comments and Suggestions for Authors
The authors have answers all my concerns.
Author Response
Thank you.
Reviewer 3 Report
Comments and Suggestions for Authors
Unfortunately, authors are unable to fully answer to the points 1, 5, 6, 7 from the first round of review due to the financial reasons. However, questions which were addresed for the authors were important and it is required to answer to these questions (at least for points 1 and 5). Moreover, authors did not answered how they checked the fact of coitus (point #2). And did not performed required analysis from the point #10.
I recommend for the authors to find additional support and reperform required experiments. Therefore, I still put Major revision.
Author Response
Dear Reviewer,
We are very sorry that we could not resolve your concerns. You can be sure that we will definitely take your valuable contributions into consideration in future studies. However, conducting animal studies in our country requires a lot of cost. We can do it with our own means. Of course, repeating the experiment would be great in line with your valuable contributions. Unfortunately, we do not have the support to provide this. 1 US Dollar = 35.86 Turkish Lira. Animal fees, Kits, animal care fees, medicines are all covered by us. I hope you will understand us. I respect your decision.
Answer: Moreover, authors did not answered how they checked the fact of coitus (point #2).
Response: In the evaluation of reproductive functions, the focus was on pregnancy formation. Due to technical difficulties, pregnancy formation was determined by daily physical examination rather than ultrasound. It would have been good to have had vaginal smear and vaginal plate follow-up during this process. However, we did not include this in the experimental design of our study. Since we could not repeat the experiment, we could not eliminate this deficiency. But you can be sure that we will take this valuable suggestion into consideration in our future studies.
Answer: 10. Lines 329 and 343. What do you mean "semi-quantitively"? You should perform analysis in concrete data. For example, you should measure the square of haemorraghy and calculate the percentage to the total square of histological section. The same for other parameters.
Response: In the first revision, we realised that we had not understood your suggestion correctly. Although it was late, the histopathological data have been converted to numerical data in accordance with your suggestion. The method of analysis was carried out using the method described for numerical data. The results and discussion in the "Material-Methods" section have been reviewed in the light of the new analysis data and modifications have been made. A new paragraph added in "Material-Methods" section. Table 2 has been revised. ‘Six random sections from each ovary were used to determine the level of histopathological damage. Each tissue section was evaluated for stromal hemorrhage, follicular hemorrhage, follicular degeneration, and edema. To determine the level of damage, the damage square was measured for each parameter and the percentage ratio to the total square of the histological section was calculated. For stromal hemorrhage and edema, <20% was considered mild damage, 20-40% moderate damage, 40-60% severe damage, and >60% very severe damage. For follicular hemorrhage and degeneration, <25% mild damage, 25-50% moderate damage, 50-75% severe damage, and >75% very severe damage were evaluated. Each tissue section was scored semi-quantitatively for stromal hemorrhage, follicular hemorrhage, follicular degeneration, and edema as absent (0), mild (1), moderate (2), severe (3) and very severe (4). Histopathological evaluation was The scoring criteria were determined by modifying the histopathological evaluation criteria of Karateke et al. 55.’

Round 3
Reviewer 3 Report
Comments and Suggestions for Authors
Can be accepted.